# Post-Traumatic Stress as a Psychological Effect of Mild Head Injuries in Children

**DOI:** 10.3390/children10071115

**Published:** 2023-06-27

**Authors:** Xenophon Sinopidis, Panagiotis Kallianezos, Constantinos Petropoulos, Despoina Gkentzi, Eirini Kostopoulou, Sotirios Fouzas, Theodore Dassios, Aggeliki Vervenioti, Ageliki Karatza, Stylianos Roupakias, Antonios Panagidis, Evangelos Blevrakis, Eleni Jelastopulu

**Affiliations:** 1Department of Pediatric Surgery, School of Medicine, University of Patras, 26504 Patras, Greece; 2Department of Pediatric Surgery, Pediatric Hospital of Patras, 26331 Patras, Greeceantpanagidis@yahoo.co.uk (A.P.); 3Department of Mathematics, University of Patras, 26504 Patras, Greece; 4Department of Pediatrics, School of Medicine, University of Patras, 26504 Patras, Greece; 5Department of Pediatric Surgery, School of Medicine, University of Crete, 71500 Heraklion, Greece; 6Department of Public Health, School of Medicine, University of Patras, 26504 Patras, Greece

**Keywords:** mild head injury, traumatic brain injury, post-traumatic stress, CTSQ, CRIES-13, children, PTSD

## Abstract

Background: Head trauma is one of the most common pediatric emergencies. While the psychological effects of severe head injuries are well studied, the psychological consequences of mild head injuries often go overlooked. Head injuries with a Glasgow Coma Scale score of 13–15, with symptoms such as headache, vomiting, brief loss of consciousness, transient amnesia, and absence of focal neurological signs, are defined as mild. The aim of this study is to evaluate the stress of children with mild head injuries and their parents’ relevant perception during the early post-traumatic period. Methods: This is a prospective cross-sectional study on a cohort of children with mild head injuries and their parents. Two questionnaires were implemented, the Child Trauma Screening Questionnaire (CTSQ) which was compiled by the children, and the Children’s Revised Impact of Event Scale (CRIES-13), compiled by their parents. Both questionnaires are widely used and reliable. The first presents an excellent predictive ability in children with a risk of post-traumatic stress disorder, while the second is a weighted self-completed detecting instrument for the measurement of post-traumatic stress in children and adolescents, with a detailed evaluation of their reactions to the traumatic incident. The participants responded one week and one month after the traumatic event. Results: A total of 175 children aged 6–14 years and 174 parents participated in the study. Stress was diagnosed in 33.7% of children after one week, and in 9.9% after one month. Parental responses suggesting stress presence in their children were 19.0% and 3.9%, respectively. These outcomes showed that mild head injuries are not so innocent. They are often underestimated by their parents and may generate a psychological burden to the children during the early post-traumatic period. Conclusions: Mild head injuries may affect the emotional welfare of children. Healthcare providers should understand the importance of the psychological effect of this overlooked type of injury. They should be trained in the psychological effect of trauma and be aware of this probability, promptly notify the parents accordingly, and provide psychological assistance beyond medical treatment. Follow-up and support are needed to avoid the possibility of future post-traumatic stress disorder. More extensive research is needed as the outcomes of this study regarded a limited population in numbers, age, and survey period. Furthermore, many children with mild head injuries do not ever visit the emergency department and stay at home unrecorded. Community-based research on the topic should therefore be considered.

## 1. Introduction

Advancement in pediatric trauma diagnosis and treatment is characterized by an explosion in evidence [1]. Evolution in critical emergency interventions, such as cardiopulmonary resuscitation and transfusion updates, provides us with data, the interpretation of which has great impact on the management of children in emergency departments [1]. Nevertheless, regardless of the enormous progress, injuries in childhood and adolescence remain a primary cause of hospitalization worldwide [2,3].

Traumatic brain injury is a major factor of morbidity and mortality, and a common cause of pediatric admissions to the emergency departments annually. It is acknowledged as the leading cause of death and morbidity among children and teenagers [4,5]. In developed countries, pediatric trauma mortality represents more than half of all childhood deaths [5,6]. Each year, 37,200 children in the United States of America sustain severe brain trauma [7]. The total annual pediatric brain trauma incidence has been estimated at 475,000 [8]. Most traumatic brain injuries (80–90%) are of low severity, with a Glasgow Coma Scale (GCS) score of 13–15 [9,10]. A tool of international acceptance since its creation in 1974, initially known as Coma Index, GCS has been established as a common language between healthcare providers. Evaluating eye opening and verbal and motor responses, with a minimum score of one point for each of the three variables, a score of 13–15 reflects a fairly good level of awareness [11].

Severe head injuries are known to cause psychological trauma and mental health dysregulation [12]. Post-traumatic stress disorder (PTSD) is an outcome that affects a lifetime and has been studied in detail [13]. PTSD may develop following exposure to an extremely threatening or horrific event or series of events. It is characterized by re-experiencing traumatic events in the present, in the form of vivid intrusive memories, flashbacks, or nightmares. It is also characterized by avoidance of thoughts and memories of the event, or avoidance of activities, situations, or people reminiscent of the event. Finally, there is persistent perception of heightened current threat, for example as indicated by hypervigilance, or an enhanced startle reaction to stimuli such as unexpected noises. The symptoms persist for several weeks at least, and cause significant impairment in personal, family, social, educational, and other important areas of functioning [14].

The likelihood of post-traumatic stress in patients with mild head injuries with a short visit to the emergency departments is less studied compared to more severe ones [15]. Clinicians who face pediatric head trauma are trained to treat and prevent neurological complications and to provide excellent clinical support to the child. Nevertheless, recognition of stress symptoms during the post-traumatic period, and identification of children who might be at risk of future PTSD, are frequently ignored.

The term mild head injury includes any head trauma with a GCS score of 13–15 on admission, occurring during at most 24 h before examination, with symptoms such as headache, vomiting, brief loss of consciousness, transient amnesia, and absence of focal neurological signs [16]. It is of note that while a variety of studies focus on pediatric trauma, only a small number of them discuss minor head injuries. If we performed a survey on the timeline of published articles in the PubMed database, using the key words [mild head injury] and [children], we should notice that since the onset of the last decade, approximately 100 articles on mild head injuries have been published annually. Only a small amount of these articles studied their psychological impact. 

The aim of the present study is to evaluate the presence of stress symptoms after mild head injuries in children, examine the risk of future PTSD, identify the possible associated risk factors that may affect the outcome, and study the parental perception of their children’s stress. In a way, it aims to contribute to the reduction in the research gap on this type of head trauma, which, in our view, has been underrated.

## 2. Materials and Methods

The stress of children following mild head injuries and the relevant perception of their parents were evaluated in this prospective cross-sectional cohort study. According to the World Health Organization, stress is defined as a state of worry or mental tension caused by a difficult situation. A range of emotions, including anxiety and irritability, is prominent [17]. The inclusion criteria for the participation in the study were: age from 6 to 14 years, fluent knowledge of Greek language, absence of neurodevelopmental issues, no recent divorce or death in the family, available pharmaceutical history, exclusion of abuse, no history of previous hospitalization for major head trauma or serious disease, hospital stay after the traumatic incident for 24 h at most, written parental consent, and child assent. The rationale for this selection was to select children who did not present other possible causes of stress at the time of their visit to the emergency department, intending to focus on the traumatic incident as the unique cause of disarrangement. Fluent Greek knowledge was considered essential to achieve optimal understanding of the questions and avoid misleading responses.

A member of the research team (P.K.), who was a professional in the emergency department, made the acquaintance with the family and explained the project of the survey. Sampling was random. If he was present at the time of admission, the contact was immediate. If not, he performed the acquaintance and discussion with the child and the escorting parent during their short stay in hospital. He explained the aims of the study and provided them with the questionnaires which they should compile and return on the designated time of one week and one month after the traumatic incident. Both parents and children agreed to participate in the survey and were made aware that they may withdraw deliberately at any time. At that time, a consent form was acquired, and demographic and clinical information were recorded. During the post-traumatic period, the same researcher kept contact with the families, either in person or with telephone calls. Prior to the deadlines of one week and one month after the traumatic incident, he reminded the participants to return the questionnaires in a timely manner.

Two questionnaires, the Child Trauma Screening Questionnaire (CTSQ) [18] and the revised Children’s Impact of Event Scale (CRIES-13) [19], were used. They were translated in Greek and underwent linguistic and psychometric validation. Both questionnaires were identical with their previously published forms in English. Correspondence with their creators was performed, to inform them of their use in our study. Both instruments are widely used, easy to compile, and reliable. CTSQ presents an excellent predictive ability in children with a risk of post-traumatic stress disorder [20]. CRIES-13 is a weighted self-completed detecting instrument for the measurement of post-traumatic stress in children and adolescents, with a detailed evaluation of their reactions to traumatic incidents [19].

The CTSQ was completed by the injured children. It identifies the symptoms of children at risk of PTSD [18]. It is considered an optimal screening test for children during the early post-traumatic period and was constructed for their timely support [18,21]. The questions are adapted to be more comprehensible for the pediatric population. The instrument evaluates two dimensions, re-experiencing (questions 1, 2, 3, 4, and 7) and hyperarousal (questions 5, 6, 8, 9, and 10). Questions describing avoidance are not included, as the creators of the questionnaire considered that symptoms of this dimension might not be apprehended well by children in the acute post-traumatic phase [18]. The assessment is dichotomous, with either positive (one point) or negative (no points) responses, depending on the experience of anxiety symptoms at the defined time after the injury. The final score ranges from 0 to 10, with values ≥5 indicating a strong association with stress and risk of PTSD [18].

CRIES-13 is widely used for the evaluation of PTSD, focusing on the reaction to the traumatic incident [19]. It was completed by the parents, as an assessment tool of their own perception of their child’s stress. It was initially constructed with eight questions as CRIES-8 [22], and it has been developed to a 13-question instrument [19]. The 13 questions examine three dimensions, i.e., arousal (questions 3, 5, 11, 12, and 13), which investigates persistent symptoms of anxiety or increased irritability; intrusion (questions 1, 4, 8, 9), which evaluates their psychological discomfort or fear when they remember the incidence; and avoidance (questions 2, 6, 7, and 10), regarding expressions of avoidance of stimuli associated with the injury. The score presents a nonlinear ascending pattern from 0 to 5 (0 = not at all, 1 = rarely, 3 = occasionally, 5 = frequently), with a range from 0 to 65. A total score ≥ 30 is indicative of moderate or serious impact of the traumatic incidence [23,24].

The extracted data were stored in Microsoft Excel templates. Each column corresponded to a specific variable. Each line contained the variable values for a single child. A coding system was created for the numerical and scaled variables, with the codes for each variable installed in a separate file. All data were coded numerically and analyzed. In some cases, grouping of categorical values data was performed.

Exploratory factor analysis (EFA) and confirmatory factor analysis (CFA) were performed for the psychometric validation of the questionnaires. To identify the relationship between stress and demographics and injury characteristics, we performed a regression analysis. We chose nonparametric techniques such as McNemar’s and Wilcoxon sign rank test to compare stress evolution one week after injury and one month later, because of the lack of normality in our data. Chi square and Fisher’s exact tests were used for categorical data. Statistical analysis was performed using IBM SPSS version 25 software (IBM Corp., Armonk, NY, USA). The threshold for statistical significance was defined as *p* < 0.05.

## 3. Results

### 3.1. Demographics and General Outcomes

A total of 377 children between 6 and 14 years old were admitted to the Children’s Hospital of Patras because of a mild head injury during a period of one year (from February 2017 to February 2018). The children stayed in hospital for a follow-up of 24 h. Random sampling resulted in the initial contact with 235 children and their parents, to whom the two questionnaires (CTSQ and CRIES-13) were provided. Of them, 175 children (response rate 75.0%) returned the questionnaires one week later, and 131 after one month. A total of 174 parents submitted completed questionnaires one week and 133 one month after the incident. The reduced response rate after the first month was attributed to the moderate nature of the studied health problem. Certainly, many participants might be more cooperative if the addressed health trouble was more serious, i.e., more severe forms of trauma.

The 175 children who participated in the survey at the end of the first week included 114 males (65.1%) aged 6–14 years (median 9.32 years), and 61 females (34.9%) of equal age range (median 9.49 years) (Table 1). The age groups of the participants included 81 (46.3%) children between 6 and 8 years old, 51 (29.1%) between 9 and 11 years old, and 43 (24.6%) between 12 and 14 years old. They were residents of both urban (*n* = 152, 86.9%) and rural (*n* = 23, 13.1%) areas. About half of the traumatic incidents occurred when an adult was present (*n* = 94, 53.7%). The most common places where trauma occurred were street (*n* = 50, 28.6%), school (*n* = 47, 26.9%), home (*n* = 36, 20.6%), and playground (*n* = 29, 16.6%). Most traumatic incidents occurred during working days (*n* = 143, 81.7%) and in the daytime (*n* = 158, 90.3%). Regarding the traumatic mechanism, most children (*n* = 122, 70.1%) had fallen onto the ground, had a collision with another child (*n* = 29, 16.7%), or were involved in a car accident (*n* = 13, 7.5%). Closed-head trauma was the most common injury (*n* = 105, 60%), and pain was the most common symptom (*n* = 127, 72.6%).

Most children arrived in their parents’ vehicle (*n* = 143, 81.7%), while 32 (18.3%) were transported by an ambulance. The majority arrived from a distance less than 20 km away (*n* = 117, 66.9%). Most admitted that they were responsible for the traumatic incident (119, 68%), and 52 (29.7%) blamed others. There was seasonal variation; most injuries occurred equally during summer (*n* = 64, 36.6%) and autumn (*n* = 64, 36.6%), while winter (*n* = 10, 5.7%) was the quietest season. On admission, all children were submitted for cranial plain radiography, while 5 (2.9%) were submitted to cranial computed tomography. Regarding the parents, 93 (52.8%) were high school graduates, 28 (20.3%) were unemployed, and 106 (60.2%) had an annual income under EUR 6000.

### 3.2. Psychometric Validation of the Questionnaires

The translated questionnaires presented good internal consistency, with a Cronbach’s alpha coefficient of 0.663 for the ten items of the CTSQ and 0.790 for the 13 items of CRIES-13 (Table 2). EFA confirmed the factor structure of the original tools. CFA showed a satisfactory fit for the factor models.

EFA analysis for CTSQ showed that the selection of two factors explained 41% of the total variability. The first explained 21.7% of the variability and had a value of 2.7, including five questions (re-experiencing). The second explained 40.9% of variability, had a value of 1.4, and included five questions as well (hyperarousal). There was a satisfactory adaptation for the model of two factors with a Kaiser–Meyer–Olkin (KMO) outcome of 0.692 (*p* < 0.001). The CFA models were one of two interrelated factors (re-experiencing and hyperarousal), and one for the existence of a possible third factor. The selected model described the best questionnaire structure, with a comparative fit index (CFI) = 0.81, Tucker–Lewis index (TLI) = 0.70, root mean square error of approximation (RMSEA) = 0.08, and degrees of freedom (df) = 2.192, χ^2^ = 74.520, *p* < 0.001.

The EFA analysis for CRIES-13 showed the presence of three factors (intrusion, avoidance, arousal) (KMO = 0.748, *p* < 0.001) and included 52.7% of the total variability. It showed that Bartlett’s test of sphericity was statistically significant (*p* < 0.001). CFA showed that the model presented significantly better adaptation, every question charged over 0.30 for the dimension for which it was destined, and there was not any significant charge of a question over a single factor (CFI = 0.83, TLI = 0.76, RMSEA = 0.09, df = 2.545, χ^2^ = 157.782, *p* < 0.001).

The analysis outcomes reported herein showed that the translation of both questionnaires presented psychological components in a similar way and level of importance as in the instruments constructed by their creators in English. Thus, they can be considered identical to the originals.

### 3.3. CTSQ Outcomes One Week after the Traumatic Incident

According to the responses to the CTSQ at the end of the first week after the traumatic incident, it was shown that 59 (33.7%) of the children presented scores ≥5. Awareness for danger was the most common response (question 9, *n* = 154, 88.5%), as opposed to lack of attention, which was the most uncommon (question 8, *n* = 9, 5.1%) (Table 3). The group of symptoms regarding re-experiencing prevailed over that of hyperarousal, implying that children suffered more from negative thoughts, memories, and reminders, rather than being at an intense status.

Boys presented lower stress scores (*p* = 0.013) compared to girls. Multiple lesions (*p* = 0.030), car accidents (*p* = 0.001), and the distribution of injuries to the face and especially on the mouth (*p* = 0.043) presented a significant association with post-traumatic stress (Table 1). At the same time, the age (*p* = 0.769), nationality (*p* = 0.569), time (*p* = 0.786), and season (*p* = 0.472) of the traumatic incident, and distance from hospital (*p* = 0.652), were statistically insignificant for the presentation of stress.

Regression analysis, performed to identify the risk factors for stress, presented a statistically significant model (χ^2^ = 39.728, *p* = 0.004). It showed that the risk to presenting stress was 2.5 times greater for girls compared to boys (*p* = 0.016), 5 times greater for children who lived in urban areas compared to those who lived in rural areas (*p* = 0.040), 4.5 times greater for children who were injured during weekends compared to workdays (*p* = 0.019), and 7 times greater for children injured at school compared to those injured at home (*p* = 0.004).

### 3.4. CTSQ Outcomes One Month after the Traumatic Incident

Outcomes were better regarding stress one month after the traumatic incident; all symptoms presented lower scores. However, 13 (9.9%) of the 131 participants presented scores in the range of stress (Table 3). Among the symptoms, awareness of potential dangers was persistently the most common symptom (question 9, *n* = 82, 62.6%). Furthermore, there was a shift from re-experiencing to hyperarousal, as there were more symptoms of this dimension (Table 3). This differentiation might be explained by the fact that during the early period after the traumatic incident, recent memories affected the children’s mental status, resulting in re-experiencing symptoms. After one month, when details faded out, hyperarousal symptoms prevailed. A possible component that aggravated this shift might be parental anxiety, which was certainly expressed with interventions of overprotection and hypervigilance of their child’s routine.

Involvement of the face (*p* = 0.016), the mechanism of injury (*p* = 0.045), and a history of open injury (*p* = 0.036) were significantly associated with stress presence (Table 1). Gender (*p* = 0.197), age (*p* = 0.807), residency (*p* = 0.639), and the presence of an escorting adult (*p* = 0.188) were not statistically significant. Herein, regression analysis did not identify any risk factors.

### 3.5. CRIES-13 Outcomes One Week after the Traumatic Incident

A total of 174 parents (47 males and 127 females, age range 24–45 years) completed the revised CRIES-13 questionnaire one week after the traumatic incident. The outcomes, which reflected their perception of their children’s post-traumatic stress, are shown in Table 4. A total of 19% evaluated that their child presented stress (score ≥ 30) one week after the injury. Question 1, describing the intrusion of the incident in children with persistent and disturbing thoughts when the facts were recalled in memory, showed greater scores (64.8%) (Table 4). Questions 3 and 13, which describe symptoms of excessive vigilance and arousal, sleep disorders, and lack of concentration, presented lower scores (Table 4).

Parents with financial problems, i.e., lower income (*p* = 0.046) or unemployed (*p* = 0.041), more frequently expressed the perception that their children presented stress. Age (*p* = 0.256), gender (*p* = 0.691), family social status (*p* = 0.526), number of children (*p* = 0.487), residency (*p* = 0.202), nationality (*p* = 0.082), educational level (*p* = 0.388), profession (*p* = 0.047), history of previous hospitalization (*p* = 0.215), and presence at the traumatic incidence (*p* = 0.649) did not affect the parental perception of stress presence in their children.

### 3.6. CRIES-13 Outcomes One Month after the Traumatic Incident

Of the 133 parents who returned completed their CRIES-13 questionnaires, the perception of stress involved only 5 (3.9%) parents. They believed that their child was thinking of the injury (15%), presented lack of concentration (4.5%), had bad dreams (5.3%), and presented hyperarousal symptoms connected to injury (33.9%) (Table 5). None of the other variables under study presented a statistically significant association with the presentation of symptoms, i.e., gender (*p* = 0.107), age (*p* = 0.580), family status (*p* = 0.509), number of children in family (*p* = 0.124), residency (*p* = 0.424), education (*p* = 0.328), profession (*p* = 0.328), and income (*p* = 0.076). There were not any risk factors identified either.

### 3.7. Stress Outcomes Evolution

McNemar’s test showed that there was a positive change with time, as more children with stress became stress relieved compared to those who became stressed while they were not stressed previously (χ^2^ = 24.735, *p* < 0.01). The Wilcoxon-signed rank test, which evaluated the median values of the CTSQ scores between the two dates, verified this outcome. The tests showed that two of the children who did not have stress in the first week presented stress after one month. Accordingly, of the children who presented stress in the first week, 32 did not have stress after one month. The two methods resulted in analogous outcomes of the CRIES-13 scores, indicating a positive change in the parental perception on their children’s behavior from the estimation of presence of stress at the first week, to absence of stress at the first month.

### 3.8. Children’s Stress and Parental Perception

Discrepancy was identified between stress presence in children and parental perception (*p* < 0.001). A total of 141 (81.0%) parents evaluated correctly that their child did not present stress. However, 23.4% of them estimated erroneously that their child did not have stress, whereas they did. On the other hand, of the 33 (19.0%) parents who identified the presence of stress in their children, 24.2% erroneously believed that they did, whereas they did not. The outcome showed that one out of five parents erroneously believed that their children either had or did not have stress.

We recently evaluated the levels of anxiety and depression in a cohort of 163 parents of the same study population, with the use of Hospital Anxiety and Depression Scale (HADS) [25]. Our findings showed that clinical grades of anxiety and depression were found in more than half of parents who escorted their children to hospital for a mild head injury [25].

Comparison of the results of the present study with those of our previous one [25] showed that there was a statistically significant association between CTSQ and the HADS outcomes regarding parental anxiety (*p* = 0.001) (Table 6). Specifically, of the 73 parents with anxiety, 19.2% of their children presented post-traumatic stress. It is of note that 49% of the children of the 51 parents with severe anxiety presented stress. There was not any association between CTSQ outcomes and depression of parents (*p* = 0.235).

There was also a statistically significant association between the outcomes of parental anxiety and the perception of their children’s stress with CRIES-13 (*p* = 0.001) (Table 7). Of the 51 parents with severe anxiety, 35.3% believed that their children presented post-traumatic stress, while of those without anxiety, only 9.6% believed that their children presented stress.

Analogous were the outcomes of association of the parental depression with the perception of their children’s stress (*p* = 0.040) (Table 8). Of the 77 who did not present depression, only 10.4% reported that their children presented stress. Parents with moderate and severe depression believed that their children had stress in 26.5% and 25% of cases, respectively.

## 4. Discussion

In this study, we investigated whether there is a psychological impact following mild head injuries in a cohort of children, and the relevant perception of their parents. It was found that there was stress in 33.7% of the children one week after the traumatic incident, while only 19.0% of their parents considered that their children had stress. Things appeared better one month later; however, 9.9% of children presented stress, while the corresponding parental perception was only 3.9%. These results might be indicative of a risk of PTSD development in the future. The outcomes imply that mild head injuries, which comprise the most common type of head trauma, might not be so innocent for certain children.

The outcomes of analogous studies are indicative of psychological implications after mild head injuries. Occasionally, these symptoms present endurance and are described with the terms PTSD [26] or post-concussion syndrome [27] when they persist beyond one month after the traumatic incident. It has been reported that 20% of children with mild traumatic brain injury present acute cognitive, physical, and psychological symptoms [26]. In a recent study, epilepsy was encountered in children with mild traumatic head injuries, with loss of consciousness or amnesia lasting under 30 min, via normal computed tomography and electroencephalogram [28]. These symptoms may persist for more than one month in 12–30% of children, and sometimes even for more than one year [29]. Stress symptoms were reported in 16% of children two weeks after head trauma, 10–12% after one month, and 6% after three months [30,31]. A study on a group of children involved in car accidents, including all grades of traumatic brain injury, showed that 38% developed PTSD after one month, and 15% after six months [32]. A meta-analysis including 72 articles showed an overall PTSD outcome of 16% in children and adolescents [33]. A study on children with mild head injuries in the community showed their association with behavioral problems such as withdrawal, emotional reactivity, aggression, and internalization [27]. Our results, which included approximately 30% stress presentation after one week, and 10% after a month following the traumatic incidence, do not differ significantly from the reported relevant studies which presented a roughly estimated range of 10–30%.

The variety in reported stress incidence after brain traumatic injuries is associated with the method of measurement. There are many criteria and instruments, and the different cut-off values set by researchers affect the outcomes [31]. Each research team focuses on particular characteristics of their study population, contributing to the variability in the methodology. Mild head injury outcomes are also affected by the fact that many patients stay at home without their children being diagnosed and recorded [34]. Furthermore, there are factors [26] which have been associated with persistent post-traumatic syndrome, such as depressive or anxiety disorder prior to the injury [35,36], parental distress [21], cognitive ability [37], younger age [38], coping strategies [39], exaggeration aiming for benefits [40], and postinjury management [41]. Even hospital experience in the form of exposure to scary stimuli such as beeping alarms or rush of staff have been proposed as contributing factors to anxiety in children [42]. All these factors may affect the outcomes of research on post-traumatic stress.

Discrepancy between the parental perception of their children’s stress and reality was identified in our study. There were parents who did not recognize the presence of stress. Conversely, there were parents who erroneously considered that their children presented stress. It is a fact that parental interpretation is influenced by their own intrinsic stress, as they project their own problems and concerns onto their children. We recently showed in our previous [25] as well as the present study that anxiety of parents was associated with the presence of stress in their children and affected their perception they had about their children’s psychological condition. Furthermore, their depression affected their own perception. Therefore, it has been suggested that researchers should pursue the child’s direct self-evaluation instead, which should be considered more significant than the parental perception [43,44,45]. Parents should improve their capacity to comprehend the potential impact of injury exposure on the child’s psychological functioning [43,44,45]. The impact of the parents’ coping style should also be examined [26]. Parental stress and associated behaviors, such as avoidance, influence children’s stress reactions [46].

An important parental issue was that of low income or unemployment. Our findings showed that these parents presented a greater perception of stress in their children after trauma. Analogous outcomes were mentioned in the literature, implicating a strong correlation between low income and anxiety [47,48]. Work satisfaction provides well-being and personal satisfaction, as daily needs are covered. The opposite is associated with negative somatic and psychical outcomes [49], among them an undermined communication between parents and children [50], affected by their own financially induced anxiety. As opposed to unemployment, the parents’ profession, educational level, and presence at the traumatic event were statistically insignificant in their perception of their children’s stress in our study.

Female gender, urban residence, and occurrence of the injury during the weekend or at school were identified as risk factors, associated with increased probability of stress presentation in the children of our study during the first week after the traumatic residence. It has been reported that girls are more prone to stress compared to boys [51]. It has also been reported that females develop PTSD about two times more than males do, which is attributed both to psychosocial and biological factors, such as estrogen concentrations [52,53]. Urban life, which is combined with less physical activity, less healthy nutrition, and more obesity, is recognized as a stress-increasing factor compared to a rural lifestyle, which provides a positive effect with relatively low stress [54]. This finding is in accordance with our study, which outlined urban residency as a risk factor for post-traumatic stress. As for children who were injured at school or on the street, they had in common that the traumatic incidence occurred away from the safety of home, which may intensify the stress reaction [55,56]. This was confirmed by our regression analysis as well. It is of note that age, nationality, time of the day, season, distance from the hospital, and the presence of an escorting adult were not associated with presence of stress in the present study. It is also of note that after one month, gender and residency were not statistically significantly associated with stress anymore, implying a more acute and sentimental initial reaction from girls and children who live in cities in the earlier post-traumatic phase.

While mild head injuries are not associated with significant physical symptoms, unlike the most severe types of head trauma, their psychological implications constitute a fairly serious outcome from the public health perspective [57]. They may modify the way of life of both children and their families [58]. Anxiety and depression may affect their parents as well and are associated with their children’s stress and their own perception [25]. Given that children with preinjury anxiety are most likely to experience persistent post-traumatic syndrome and anxiety, predisposing children to developing significant stress responses after a potentially traumatic event [26], it is of high importance to distinguish the children who already experienced anxiety from those who experienced post-traumatic stress with the mild head injury as the main trigger. A vicious negative feedback cycle may prolong and exacerbate this phenomenon both for children and their families [59]. The outcome is that the stress does not end in the emergency department, but continues at home, where the parent’s psychological flexibility may act as a protective factor against the development of the child’s mental distress [60]. However, as shown herein, this does not always occur.

## 5. Limitations

The study has limitations. It included children of a confined age range, followed up for a limited period of one month. We estimated that it would be difficult to follow up the already reduced pool of responding participants for more months. We experienced that, because of the mild character of the health deterioration of their children, many parents became reluctant to comply with our requests. Nevertheless, this short follow-up period is a technical limitation. We also attributed the reduction in respondents after only one month after the traumatic incident to the moderate nature of the health problem. An additional limitation is that the measured stress might also be related to the short stay in the hospital, and it may not only reflect the occurrence of the brain injury. Furthermore, many children with a mild head injury either do not ever visit the emergency department or they return home immediately after clinical and radiological examination. Therefore, more extended research is needed for the extraction of conclusions of greater validity. Community-based research on the topic should also be considered.

## 6. Conclusions

The study outcomes showed that even mild head injuries may induce persistent psychological symptoms. In the early post-traumatic period, a proportion of 33.7% of children presented stress, while at the end of the first month, the percentage was reduced to 9.9%. Their parents presented a perception of their stress in 19.0% of cases after the first week and 3.9% after one month. This perception was erroneous in 23.4% of parents who believed that their child did not have stress while they did, and in 24.2% who believed that they had stress while they did not. We also found that the presence of stress and the perception of the parents were affected by the parental symptoms of anxiety and depression.

There are three factors that we believe that future research should emphasize. The first regards the clear definition of each psychological symptom and combination of symptoms, and attachment to the suggestions of definitions set by organizations such as the International Statistical Classification of Diseases and Related Health Problems (ICD). It is the safest way to obtain uniformity in the description and evaluation of health deviations and create comparable outcomes. The second regards the isolation of possible factors that might affect the presence and endurance of psychological issues after mild head injuries. These should be investigated in children as well, but mostly in their parents. With this maneuver, a clearer representation of the true incentives of psychological issues will be obtained. Finally, a more psychologically oriented education of clinical practice should be organized by medical schools, especially during the residency years. Clinicians involved in the management of pediatric trauma are trained to focus on the emergency character of neurological clinical presentation. Nevertheless, as shown herein, there is stress in a minority of children, which may persevere after a traumatic incident and, in some cases, evolve into PTSD. To limit this complication, emergency specialists, pediatricians, and pediatric surgeons should receive not only clinical but also psychology training to support both children and parents. Furthermore, healthcare systems should provide continued outpatient follow-up with specialists who may support, consult, and identify the children who are at increased risk of stress presentation.

## Figures and Tables

**Table 1 children-10-01115-t001:** Demographics and most important injury characteristics of children who presented stress (CTSQ score ≥ 5) one week and one month after the traumatic incident.

Demographics and Injury Characteristics	One Week after Injury (*n* = 175)				One Month after Injury (*n* = 131)			
	Stress		No Stress		Stress		No Stress	
Variable	*n*	%	*n*	%	*n*	%	*n*	%
Gender								
Male	31	27.2	83	72.8	6	7.3	76	92.7
Female	28	45.9	33	54.1	7	14.3	42	85.7
Age groups								
6–8	27	33.3	54	66.7	6	10.5	51	89.5
9–11	19	37.3	32	62.7	3	7.5	37	92.5
12–14	13	30.2	30	69.8	4	11.8	30	88.2
Residence								
Urban	53	34.9	99	65.1	11	9.5	105	90.5
Rural	6	26.1	17	73.9	2	13.3	13	86.7
Adult supervision								
Yes	39	41.5	55	58.5	59	86.8	9	13.2
No	20	24.7	61	75.3	59	93.7	4	6.3
Location of injury								
School	16	34.0	31	66.0	0	0.0	30	100.0
Home	6	16.7	30	83.3	0	0.0	30	100.0
Street	27	54.0	23	46.0	9	23.1	30	76.9
Playground	6	20.7	23	79.3	3	12.5	21	87.5
Other	4	30.8	9	69.2	1	12.5	7	87.5
Injury mechanism								
Fall on the ground	33	56.9	89	76.7	6	50.0	83	70.3
Accidental collision with another child	5	8.6	24	20.7	1	8.3	20	16.9
Car accident	10	17.2	3	2.6	3	25.0	10	8.5
Drift by a car	3	5.2	0	0.0	1	8.3	1	0.8
Drift by a motorcycle	5	8.6	0	0.0	1	8.3	3	2.5
Quarrel with another child	1	1.7	1	0.9	0	0.0	1	0.8
Direct force impact to the head	16	27.6	52	44.8	2	16.7	50	42.4
Type of injury								
Closed-head trauma	32	54.2	47	40.5	1	7.7	28	23.7
Open-head trauma	19	32.2	51	44.0	9	69.2	46	39.0
Temporary loss of conscience	18	30.5	24	20.7	3	23.1	30	25.4
Multiple lesions	19	32.2	20	17.2	6	46.2	24	20.3
Head area involved								
Cranial vault	59	100.0	116	100.0	13	100	118	100
Face	17	28.8	16	13.8	6	46.2	21	17.8
Ocular area	3	5.1	2	1.7	0	0.0	3	2.5
Nose	2	3.4	3	2.6	2	15.4	3	2.5
Chin	3	5.1	2	1.7	3	23.1	2	1.7
Mouth	5	8.5	1	0.9	3	23.1	2	1.7
Forehead	13	22.0	24	20.7	3	23.1	22	18.6
Neck	3	5.1	4	4.3	1	7.7	4	3.4

**Table 2 children-10-01115-t002:** Internal consistency of the two questionnaires.

Variable	Cronbach’s Alpha	Cronbach’s Alpha Based on Standardized Items	*n* of Items
CTSQ	0.663	0.638	10
CRIES-13	0.790	0.795	13
CRIES-13-Intrusion	0.490	0.493	4
CRIES-13-Avoidance	0.717	0.727	4
CRIES-13-Arousal	0.885	0.596	5

**Table 3 children-10-01115-t003:** CTSQ outcomes one week and one month after the traumatic incident. Dimensions evaluated: re-experiencing (questions 1, 2, 3, 4, and 7) and hyperarousal (questions 5, 6, 8, 9, and 10).

CTSQ Questions	Positive Responses after One Week (*n* = 175)		Positive Responses after One Month (*n* = 131)	
	*n*	%	*n*	%
1. Do you have thoughts or memories about the accident that you do not want to have?	102	58.3	9	6.8
2. Do you have bad dreams about the accident?	27	15.4	4	3.1
3. Do you feel or act, as if the accident is about to happen again?	64	36.6	30	22.7
4. Do you have bodily reactions (such as fast-beating heart, stomach churning, sweating, and feeling dizzy) when reminded of the accident?	56	32.0	20	15.3
5. Do you have trouble falling or staying asleep?	22	12.6	1	9.1
6. Do you feel grumpy or lose your temper?	28	16.0	19	14.4
7. Do you feel upset by reminders of the accident?	122	69.7	58	44.7
8. Do you have a tough time paying attention?	9	5.1	2	1.5
9. Are you on the “look-out” for possible dangerous things that might happen to yourself and others?	154	88.5	82	62.6
10. When things happen by surprise or all of a sudden, does it make you “jump”?	61	34.9	34	25.8

**Table 4 children-10-01115-t004:** CRIES-13 outcomes one week after the traumatic incident. Dimensions evaluated are arousal (questions 3, 5, 11, 12, and 13), intrusion (1, 4, 8, and 9), and avoidance (questions 2, 6, 7, and 10).

CRIES-13 Questions	Parental Responses (*n* = 174)							
	Not at All		Rarely		Sometimes		Often	
	*n*	%	*n*	%	*n*	%	*n*	%
1. Do you have the impression that your child has to think about it often?	31	17.6	31	17.6	76	43.2	38	21.6
2. Does your child try to put it out of his/her mind?	70	39.8	13	7.4	41	23.3	52	29.5
3. Does your child find it difficult to pay attention or concentrate?	146	83.0	9	5.1	17	9.7	4	2.3
4. Does your child have sudden surges of strong feelings?	85	48.3	32	18.2	45	25.6	14	8.0
5. Does your child get startled more easily or is she/he more nervous than before it happened?	76	43.2	28	15.9	48	27.3	24	13.6
6. Does your child stay away from things that remind him/her of the event (like places or situations)?	102	58.0	20	11.4	26	14.8	28	15.9
7. Does your child try not to talk about it?	87	49.4	9	5.1	21	11.9	59	33.5
8. Does your child suddenly see images of the event in her/his mind or have bad dreams?	145	82.4	9	5.1	13	7.4	9	5.1
9. Do other things keep making your child think of it?	56	31.8	35	19.9	71	40.3	14	8.0
10. Does your child try not to think about it?	105	59.7	22	12.5	18	10.2	31	17.6
11. Does your child get easily irritated or angry?	68	38.6	25	14.2	60	34.1	22	12.5
12. Is your child overly cautious or on guard even when there is no clear need to be?	79	44.9	24	13.6	44	25.0	29	16.5
13. Does your child have trouble sleeping?	150	85.7	7	4.0	9	5.1	9	5.1

**Table 5 children-10-01115-t005:** CRIES-13 outcomes one month after the traumatic incident. Dimensions evaluated are arousal (questions 3, 5, 11, 12, and 13), intrusion (1, 4, 8, and 9), and avoidance (questions 2, 6, 7, and 10).

CRIES-13 Questions	Parental Responses (*n* = 133)							
	Not at All		Rarely		Sometimes		Often	
	*n*	%	*n*	%	*n*	%	*n*	%
1. Do you have the impression that your child has to think about it often?	65	48.9	48	36.1	16	12.0	4	3.0
2. Does your child try to put it out of his/her mind?	103	77.4	6	4.5	13	9.8	11	8.3
3. Does your child find it difficult to pay attention or concentrate?	125	94.0	2	1.5	4	3.0	2	1.5
4. Does your child have sudden surges of strong feelings?	71	53.4	20	15.0	41	30.8	1	0.8
5. Does your child get startled more easily or is she/he more nervous than before it happened?	66	49.6	17	12.8	44	33.1	6	4.5
6. Does your child stay away from things that remind him/her of the event (like places or situations)?	96	72.2	18	13.5	14	10.5	5	3.8
7. Does your child try not to talk about it?	79	59.4	17	12.8	7	5.3	30	22.6
8. Does your child suddenly see images of the event in her/his mind or have bad dreams?	117	88.0	9	6.8	6	4.5	1	0.8
9. Do other things keep making your child think of it?	57	42.9	30	22.6	39	29.3	7	5.3
10. Does your child try not to think about it?	111	83.5	9	6.8	6	4.5	7	5.3
11. Does your child get easily irritated or angry?	53	39.8	16	12.0	60	45.1	4	3.0
12. Is your child overly cautious or on guard even when there is no clear need to be?	64	48.1	24	18.0	34	25.6	11	8,3
13. Does your child have trouble sleeping?	116	89.2	3	2.3	8	6.2	3	2.3

**Table 6 children-10-01115-t006:** Association of parental anxiety levels (HADS) and stress outcomes in children (CTSQ).

*p* = 0.001		Parental Anxiety Evaluation (HADS)		
		Low	Moderate	Severe
		*n* (%)	*n* (%)	*n* (%)
Children stress evaluation (CTSQ)	Children with stress	14 (19.2)	16 (41.0)	25 (49.0)
	Children without stress	59 (80.8)	23 (59.0)	26 (52.0)
	Total	73	39	51

**Table 7 children-10-01115-t007:** Association of parental anxiety levels (HADS) and parental perception on their children’s presence of stress (CRIES-13).

*p* = 0.001		Parental Anxiety Evaluation (HADS)		
		Low	Moderate	Severe
		*n* (%)	*n* (%)	*n* (%)
Parental perception (CRIES-13)	Children with stress	7 (9.6)	5 (13.2)	18 (35.3)
	Children without stress	66 (90.4)	33 (86.8)	33 (64.7)
	Total	73	38	51

**Table 8 children-10-01115-t008:** Association of parental depression levels (HADS) and parental perception on their children’s presence of stress (CRIES-13).

*p* = 0.040		Parental Depression Evaluation (HADS)		
		Low	Moderate	Severe
		*n* (%)	*n* (%)	*n* (%)
Parental perception (CRIES-13)	Children with stress	8 (10.4)	13 (26.5)	9 (25.0)
	Children without stress	69 (89.6)	36 (73.5)	27 (75.0)
	Total	77	49	36

## Data Availability

Not applicable.

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
