# Peer review of "Post-Traumatic Stress as a Psychological Effect of Mild Head Injuries in Children"

_children, 2023, doi:10.3390/children10071115_

Round 1
Reviewer 1 Report
Title
· Consider changing "overlooked" to a less subjective term, unless your data provides compelling evidence that this issue is commonly overlooked.
Abstract
· Explain the importance of understanding the psychological consequences of mild head injuries in children.
· Define key terms like "mild head injuries".
· Provide more detail on the use and administration of the two questionnaires.
· Offer brief interpretations of the statistics presented in the abstract.
· Mention the study's limitations and future research directions.
Introduction
· Provide a broader context before focusing on pediatric traumatic brain injuries.
· Clarify what is meant by 'mild' head injuries.
· Mention the gap in previous research that your study aims to fill.
· Explicitly state the research questions or hypotheses.
· Enhance the statement of your study's aim and how it relates to the research gap and questions.
· Explain terms and concepts like PTSD to make the introduction more accessible.
Materials and Methods
· Provide more context around terms like the Glasgow Coma Scale score and 'stress symptoms'.
· Justify your inclusion and exclusion criteria.
· Provide more detail on your sampling strategy, including subject selection and the necessity of Greek fluency.
· Explain the choice of the CTSQ and CRIES-13 questionnaires.
· Discuss how data was stored, coded, and prepared for analysis.
· Explain why you chose certain statistical tests and how they address your research questions.
Results
· Clarify your presentation of statistical results, including factor and confirmatory factor analysis.
· Explain unexpected findings, like the shift from reexperiencing to hyperarousal symptoms.
· Discuss potential biases, such as the decreased response rate from the first to the second questionnaire.
· Compare your results to previous similar studies.
· Discuss how missing data was treated.
· Justify your sample size.
· Consider presenting complex results in tables or figures for easier comprehension.
· Explain confusing statistical terms, like "stress-relieved".
· Discuss non-significant findings more thoroughly.
· Discuss why certain variables were included and others were not.
· Offer more interpretation of your findings, especially regarding parental perceptions of children's stress levels.
Discussion
· Avoid or explain technical jargon.
· Discuss the study's limitations.
· Contextualize your results within existing literature.
· Discuss parental stress and its influence on perceptions of child stress.
· Discuss non-significant findings.
· Discuss the implications of your results more thoroughly.
· Suggest directions for future research.
· Include a clear conclusion summarizing the main findings and their implications.
References
Your references appear to cover a comprehensive range of topics and sources related to pediatric traumatic brain injury, which suggests a well-researched and comprehensive study.
Phrase/Sentence Structure and Clarity: The sentences in the abstract, introduction, and methods sections could be restructured for clarity and conciseness.
For instance, the sentence:
"The psychological effects of severe head injuries are well-known and have been studied thoroughly. However, the psychological consequences of mild head injuries are often ignored."
Could be written as:
"While the psychological effects of severe head injuries are well-studied, the psychological consequences of mild head injuries often go overlooked."
Concision: Some sentences can be more concise. For example, the phrase:
"It is well known that severe head injuries are associated with psychological trauma and mental health dysregulation."
Could be:
"Severe head injuries are known to cause psychological trauma and mental health dysregulation."
Author Response
Comments and Suggestions for Authors and Authors’ Responses
Authors’ comment: We thank the reviewer for the comments. We responded to each comment accordingly. The modifications are shown in the text in characters in red color.
Title
Consider changing "overlooked" to a less subjective term, unless your data provides compelling evidence that this issue is commonly overlooked.
Response: The term “overlooked” has been eliminated to provide a definitively objective title.
Abstract
Explain the importance of understanding the psychological consequences of mild head injuries in children.
Response: We attempted to explain this concept [page 1, lines 34-39].
Define key terms like "mild head injuries".
Response: The term has been defined [page 1, lines 17-19].
Provide more detail on the use and administration of the two questionnaires.
Response: Details have been provided. [page1, lines 24-28].
Offer brief interpretations of the statistics presented in the abstract.
Response: Brief interpretation has been provided [page 1, lines 31-33].
Mention the study's limitations and future research directions.
Response: Both reviewer’s suggestions have been addressed at the end of the abstract [page 1, lines 38-42].
Introduction
Provide a broader context before focusing on pediatric traumatic brain injuries.
Response: An introductory paragraph has been added for this scope [page 2, lines 48-53].
Clarify what is meant by 'mild' head injuries.
Response: The definition of mild head injuries has been addressed [page 2, lines 83-86].
Mention the gap in previous research that your study aims to fill. Explicitly state the research questions or hypotheses. Enhance the statement of your study's aim and how it relates to the research gap and questions.
Response: The issues the reviewer addresses herein are mentioned [page 2, lines 86-91, and 95-96].
Explain terms and concepts like PTSD to make the introduction more accessible.
Response: The term PTSD has been addressed according to the ICD-11 definition [page 2, lines 68-76].
Materials and Methods
Provide more context around terms like the Glasgow Coma Scale score and 'stress symptoms'.
Response: Context has been added for both terms [page 3, lines 99-102, and page 2, lines 61-65].
Justify your inclusion and exclusion criteria.
Response: The justification of criteria has been added [page 3, lines 107-111].
Provide more detail on your sampling strategy, including subject selection and the necessity of Greek fluency.
Response: As for the Greek fluency, it has been considered essential, since our region is inhabited by multiple nationalities, tourists, and immigrants from the Balkans, Eastern Europe, Asia, and Africa, many of them presenting language difficulties. A comment was added [page 3, lines 114-116]. Details on sampling strategy and practice have been added [page 3, additions in red in lines 112-124].
Explain the choice of the CTSQ and CRIES-13 questionnaires.
Response: The choice has been explained [page 3, lines 130-133].
Discuss how data was stored, coded, and prepared for analysis.
Response: A paragraph was added with the requested information [page 4, lines 156-162].
Explain why you chose certain statistical tests and how they address your research questions.
Response: The statistical analysis was rewritten in a clearer way explaining the reasoning of the statistical tests chosen [page 4, lines 163-170].
Results
Clarify your presentation of statistical results, including factor and confirmatory factor analysis.
Response: Statistical results on EFA and CFA have been presented more clearly [pages 6 and 7, red additions in lines 214-234].
Explain unexpected findings, like the shift from reexperiencing to hyperarousal symptoms.
Response: The point has been addressed [page 8, lines 266-271].
Discuss potential biases, such as the decreased response rate from the first to the second questionnaire.
Response: We discussed the decreased response rate [page 4, lines 180-183].
Compare your results to previous similar studies.
We compared more vigorously our results to previous studies in the discussion section.
Discuss how missing data was treated.
In our data analysis we used the full completed questionnaires with no missing data. As regards to comparing stress evolution, we used the completed questionnaires filled in both time periods.
Justify your sample size.
For Factor Analysis, Kaiser-Meyer-Olkin Measure of Sampling Adequacy is near or greater to 0.7 in both cases (EFA and CFA) which is quite good according to Kaiser and Rice (1974) and no further data collection was needed. For the regression analysis we have a lower intermediate sample size. According to Long (1997), sample sizes of less than 100 should be avoided. For the nonparametric statistical techniques there is no problem at all. (References: Kaiser, H. F., & Rice, J. (1974). Little Jiffy, Mark Iv. Educational and Psychological Measurement, 34(1), 111–117, and Long, J.S. (1997). Regression Models for Categorical and Limited Dependent Variables. Thousand Oaks, CA: SAGE Publications, Inc.).
Consider presenting complex results in tables or figures for easier comprehension.
Response: We tried to explain more plainly all results that we considered quite complex (sessions of factorial analysis, stress outcomes evolution, parental perception, additions in red). We added three more tables to explain new insights of analysis (Tables 6-8).
Explain confusing statistical terms, like "stress-relieved".
Response: The confusing term was removed [page 10, lines 317-318].
Discuss non-significant findings more thoroughly.
Response: Non-significant findings were added in detail as well [page 8, lines 250-252 and 273-275, page 9, lines 293-295, and 302-304].
Discuss why certain variables were included and others were not.
We chose the most interesting and appropriate for a manuscript of limited text. However, we added more information on variables which were not discussed so far in a paragraph [page 5, lines 197-205].
Offer more interpretation of your findings, especially regarding parental perceptions of children's stress levels.
Response: We have already discussed the discrepancy between parental perception and children’s stress. In addition, we performed a comparison of the results of our previous recent study on the same population (reference 25) on the anxiety of parents with the HADS scale. We compared those results with the outcomes of CTSQ and CRIES-13 in the present study. Three more tables have been added as well [pages 10-12, lines 327-355].
Discussion
Avoid or explain technical jargon.
Response: We tried to avoid any technical difficulties for the reader, explaining each term in the text as better as possible.
Discuss the study's limitations.
Response: We discussed the study’s limitations in a separate paragraph. We added two more limitations as well [page 14, lines 456-468].
Contextualize your results within existing literature.
Response: Contextualization has been performed in discussion and has been extended [pages 12-14, lines 382-384, 402-406, 412-414, 419-421, 422-425, 430-431, 434].
Discuss parental stress and its influence on perceptions of child stress.
Response: We discussed the issues of parental stress and its influence [page 13, lines 398-421, existing text and additions in red]
Discuss non-significant findings.
Response: We added this issue [page 13, lines 434-439].
Discuss the implications of your results more thoroughly.
Response: We performed this more thoroughly in the last paragraph of the discussion section, and in the conclusions section [pages 13-14, additions in red].
Suggest directions for future research.
Response: Suggestive directions have been implied in the conclusions section [page 14, lines 478-496].
Include a clear conclusion summarizing the main findings and their implications.
Response: Outlined findings are in the first, and implications in the second part of the conclusions section [page 14, additions in red].
References
Your references appear to cover a comprehensive range of topics and sources related to pediatric traumatic brain injury, which suggests a well-researched and comprehensive study.
Response: We thank the reviewer for the kind comment. The revision of the manuscript resulted in a total of 60 references.
Comments on the Quality of English Language
Phrase/Sentence Structure and Clarity: The sentences in the abstract, introduction, and methods sections could be restructured for clarity and conciseness.
For instance, the sentence:
"The psychological effects of severe head injuries are well-known and have been studied thoroughly. However, the psychological consequences of mild head injuries are often ignored."
Could be written as:
"While the psychological effects of severe head injuries are well-studied, the psychological consequences of mild head injuries often go overlooked."
Response: The modification has been performed [page 1, lines 15-19].
Concision: Some sentences can be more concise. For example, the phrase:
"It is well known that severe head injuries are associated with psychological trauma and mental health dysregulation."
Could be:
"Severe head injuries are known to cause psychological trauma and mental health dysregulation."
Response: The modification has been performed [page 2, lines 66-67]. We also performed concision in other points of the text.
Reviewer 2 Report
Thank you for submitting your manuscript. I enjoyed reading it. The psychological impact and repercussion in children and their parents after a traumatic accident, is becoming a subject of interest. As you mention in your manuscript several factors could play an important role and sometimes could be difficult to determine which factors may affect in a higher and lower mode these psychological side effects in the different pediatric population.
In general, I found the data of this prospective cross-sectional cohort study focused in mild head trauma interesting. However, I have some questions and observations for the authors
1. In the results, you mentioned the age groups of 6-8, 9-11 and 12-14 years-old. Did you analyze statistical differences between these different groups of age regarding the amount of stress and the differences of regression in the time-line? Especially in the situation of motor-vehicle accidents and facial wounds associated.
2. In the inclusion criteria you did not mentioned other lesions associated beside the mild head trauma. Were there differences regarding the amount of injuries among these children? Did some of these children had a head RX films or CT scan?
3. The decreasing trend of the psychological stress in the children of your study was clear at 1 month. Could you explain your motivation for not controlling the trend in a longer period such as 3, 6 and 12 months? Especially if you are proposing the possibility of a persistent post-traumatic psychological disorder (PTSD) in mild head trauma. The follow-up could be too short to support this conclusion
4. Why you did not analyze also the amount of stress in the parents during this period? It is known that the psychological status of the parents could have a high probability to influence the psychological status of their children.
5. Table 4 and 5 will need to specify in a clear mode the name of each column in order to avoid misinterpretation
6. As continuity of your study, could be very interesting to compare mild head trauma versus other mild situations such as moderate burns, not severe wounds for animal bites, etc., in order to evaluate the degree of psychological impact in a time-line. This could help to elucidate if the persistence of the psychological stress could be related to the psychological experience of the accident or eventually, due to an anatomical brain lesion unrecognized in a mild head trauma.
Author Response
Comments and Suggestions for Authors and Authors’ Responses
Authors’ comment: We thank the reviewer for the comments. We responded to each comment accordingly. The modifications are shown in the text in characters in red color.
Thank you for submitting your manuscript. I enjoyed reading it. The psychological impact and repercussion in children and their parents after a traumatic accident, is becoming a subject of interest. As you mention in your manuscript several factors could play an important role and sometimes could be difficult to determine which factors may affect in a higher and lower mode these psychological side effects in the different pediatric population.
In general, I found the data of this prospective cross-sectional cohort study focused on mild head trauma interesting. However, I have some questions and observations for the authors.
- In the results, you mentioned the age groups of 6-8, 9-11 and 12-14 years old. Did you analyse statistical differences between these different groups of age regarding the amount of stress and the differences of regression in the timeline? Especially in the situation of motor-vehicle accidents and facial wounds associated.
Response: We thank the reviewer for the kind comments. We analyzed the outcomes according to the variable of age. We did not find any statistically significant differences. Therefore, we added this in page 8, lines 250 and 274.
- In the inclusion criteria you did not mention other lesions associated beside the mild head trauma. Were there differences regarding the number of injuries among these children? Did some of these children had a head RX films or CT scan?
Response: The study population included with mild head injuries only. Only minor injuries such as bruises were included. Our scope was to focus only to a cohort with mild head injury, without other health problems. All children were examined with head plain radiographies. CT was performed in 2.9% and we added this information in page 5, lines 202-204.
- The decreasing trend of the psychological stress in the children of your study was clear at 1 month. Could you explain your motivation for not controlling the trend in a longer period such as 3, 6 and 12 months? Especially if you are proposing the possibility of a persistent post-traumatic psychological disorder (PTSD) in mild head trauma. The follow-up could be too short to support this conclusion.
Response: The reviewer is right, we had to persist more in time depth. We already confronted difficulties to persuade many parents and their children to respond one month after, as only 9.9% presented stress at one month, and 3.9% perception of their parents. We attributed this to the moderate character of the trauma and discussed it as well. But because the reviewer mentioned it, we added this parameter to the limitations of the study [page 4, lines 180-183, page 14, lines 457-462].
- Why you did not also analyse the amount of stress in the parents during this period? It is known that the psychological status of the parents could have a high probability to influence the psychological status of their children.
Response: We studied anxiety and depression of the parents of the same study population and published our results in a previous publication [reference 25]. However, we added a comparison of the results of that paper with the outcomes of the present study [pages 10-12, lines 332-356].
- Table 4 and 5 will need to specify in a clear mode the name of each column in order to avoid misinterpretation.
Response: We added a row (n and %) to clarify better the two tables.
- As continuity of your study, could be very interesting to compare mild head trauma versus other mild situations such as moderate burns, not severe wounds for animal bites, etc., in order to evaluate the degree of psychological impact in a timeline. This could help to elucidate if the persistence of the psychological stress could be related to the psychological experience of the accident or eventually, due to an anatomical brain lesion unrecognized in a mild head trauma.
Response: We thank the reviewer for this proposal. We will do our best towards this scope. Furthermore, we discussed this bias in the discussion section and in the conclusions.
Round 2
Reviewer 1 Report
Significant improvements have indeed been implemented, and I am grateful for this. I derive immense pleasure from having had the opportunity to contribute to the academic growth of this document.